# Scalable Neural Data Server: A Data Recommender for Transfer Learning

**Tianshi Cao** [1,2,3] *    **Sasha Doubov** [1,2] *    **David Acuna** [1,2,3]    **Sanja Fidler** [1,2,3]

University of Toronto[1]   Vector Institute[2]   NVIDIA[3]

{jcao,doubovs,davidj,fidler}@cs.toronto.edu

## Abstract

Absence of large-scale labeled data in the practitioner's target domain can be a bottleneck to applying machine learning algorithms in practice. Transfer learning is a popular strategy for leveraging additional data to improve the downstream performance, but finding the most relevant data to transfer from can be challenging. Neural Data Server (NDS) [45], a search engine that recommends relevant data for a given downstream task, has been previously proposed to address this problem. NDS uses a mixture of experts trained on data sources to estimate similarity between each source and the downstream task. Thus, the computational cost to each user grows with the number of sources. To address these issues, we propose Scalable Neural Data Server (SNDS), a large-scale search engine that can theoretically index thousands of datasets to serve relevant ML data to end users. SNDS trains the mixture of experts on intermediary datasets during initialization, and represents both data sources and downstream tasks by their proximity to the intermediary datasets. As such, computational cost incurred by SNDS users remains fixed as new datasets are added to the server. We validate SNDS on a plethora of real world tasks and find that data recommended by SNDS improves downstream task performance over baselines. We also demonstrate the scalability of SNDS by showing its ability to select relevant data for transfer outside of the natural image setting.

## 1   Introduction

In recent years, machine learning (ML) applications have taken a foothold in many fields. Methodological and computational advances have shown a trend in performance improvements achieved by using larger and larger training datasets, confirming that data is the fuel of modern machine learning. This fuel, however, is not free of cost. While the raw data itself (such as images, text, etc.) can be relatively easy to collect, annotating the data is a labour intensive endeavour. A popular and effective approach to reduce the need of labeling in a target application domain is to leverage existing datasets via transfer learning. Transfer learning is the re-purposing of ML models trained on a source dataset towards a different target task [22]. The performance of transfer learning is predicated on the relevance of the source to the target [46, 36]. Although there are numerous datasets available through various data sharing platforms [41, 5, 38, 20], finding the right dataset that will most benefit transfer-learning performance on the target domain is not a simple problem.

Consider a startup looking to train a debris detector for use on a road cleaning robot, and suppose that they have collected a small, labeled dataset. They are looking for data with which to augment their training set. Ideally, they would find a dataset of roadside debris that is captured from a sensor similar to their own. However, keyword searches for "roadside debris", would likely find data geared towards autonomous driving, which might be from one of the many sensor types (rgb, lidar, radar). Searching for their specific sensor type might lead to data from similar setups, but of different objects.

---

*Equal contribution

35th Conference on Neural Information Processing Systems (NeurIPS 2021).

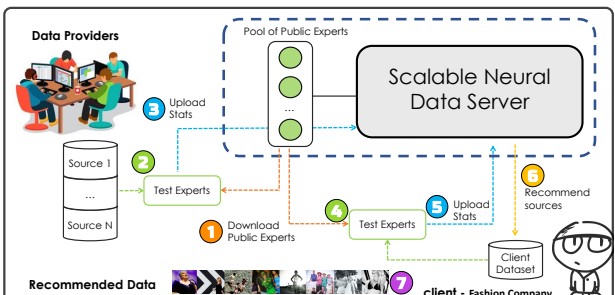

Figure 1: **Scalable Neural Data Server.** SNDS is a scalable data recommendation system for transfer learning data. It can theoretically index thousands of datasets by using a *pool of experts*, trained once on intermediary datasets, and used for data matching. SNDS uses these intermediary datasets to compute similarities between the consumer's data and the server's datasets. Similarities are computed on the user's side with a computation cost that does not grow with the size of the server's data. User's data privacy is preserved.

Rather than using keywords, an ideal system would find relevant data based on the *training set* itself. In this work, we envision a data search engine that indexes source data from data providers, and recommends relevant transfer learning data to consumers who have a target task.

Neural DataServer (NDS) [45] has been proposed as a data recommendation system for transfer learning. Given a target task (as represented by a target dataset), it uses expert models trained on splits of source data to make predictions on the target, which is in turn used as a measure of similarity between the source and the target. Directly indexing a large number of datasets with NDS is expensive, as it would require each data provider to upload a trained expert, and any data consumer looking to select data for purchase must download and run all of the trained experts in order to evaluate similarity. Hence, for $M$ data providers in the server, the computational and bandwidth cost to the data consumer scales linearly with $M$.

At the same time, using an expert model per data provider in NDS ensures that the data recommendations are highly relevant to the consumer. A data search engine that operates in a large-scale setting must be able to preserve the quality of recommendations, while disentangling the consumer's computational cost from the number of providers. The system should also preserve the data privacy properties from NDS, where the client's data is never exchanged with the server.

We propose Scalable Neural Data Server (SNDS), a search engine for data recommendation at scale. This is achieved by using an intermediary, public dataset to train a fixed number of experts, and a similarity measure that evaluates the relevance of the server's data for a given consumer via these experts. Compared to NDS, the computational cost of SNDS is constant for a given consumer with respect to the $M$ data providers, and so is the bandwidth cost between the consumer and the server. We validate SNDS experimentally and find that 1) SNDS generalizes in the sense that it recommends useful data for downstream tasks that are not directly represented by the intermediary datasets, 2) SNDS improves performance on downstream tasks over baselines, and is competitive with NDS, 3) SNDS is able to generalize to domains that are very dissimilar to the intermediary datasets.

## 2 Background

In this section, we first formalize our problem setting which is similar to that of NDS, and then provide a brief overview of the data selection method introduced in NDS.

### 2.1 Problem setting

Let $\mathcal{X}$ denote a sample space and $\mathcal{Y}$ a label space. Suppose that there are currently $M$ data providers (which we call "sources"), each with data available for use in transfer learning. Let the sources be denoted as $\mathbb{S} = \{S_1, \ldots, S_M\}$, where $S_i$ denote a set of $m_i$ sample-label pairs $((\mathbf{x}_1, \mathbf{y}_1), \ldots, (\mathbf{x}_{m_i}, \mathbf{y}_{m_i})) \in (\mathcal{X} \times \mathcal{Y})^{m_i}$. The target task by the data consumer is represented by a target specific distribution $\mathcal{D}$ supported over $\mathcal{X} \times \mathcal{Y}$, from which the target dataset $T = ((\mathbf{x}_1, \mathbf{y}_1), \ldots, (\mathbf{x}_n, \mathbf{y}_n)) \in (\mathcal{X} \times \mathcal{Y})^n$ is drawn. The goal of the data consumer is to find data sources suitable for transfer learning to their target task. Specifically, we seek a mixture of sources $S^*$ that minimizes the following risk for the data consumer:

$$R(S) = E_{\mathbf{x}, \mathbf{y} \sim \mathcal{D}}[l(h_{S \cup T}(\mathbf{x}), \mathbf{y})] \tag{1}$$

while satisfying some budget constraint of $|S| \leq b$, as specified by the data consumer. Here, $h_{S \cup T}$ is a predictive model $h : \mathcal{X} \to \mathcal{Y}$ that is trained on $S$ and $T$. In practice, $h$ is often first trained on $S$ during pretraining, and then trained on $T$ during finetuning.

**Desiderata 2.1.** *To meet the requirements of a data search engine, a potential solution should satisfy:*

- *D1: Data in $\mathbb{S}$ or $T$ are not stored on the server.*

- *D2: Data in $\mathbb{S}$ is not revealed to the consumers before $S^*$ is selected.*

- *D3: Data in $T$ is not revealed to any data provider or any other consumers throughout the process.*

- *D4: The computational and bandwidth cost for the consumer should be fixed as $M$ (i.e the number of data providers) grows.*

What makes this problem challenging is that D1-D3 prevents the sources and the target from being on the same device, and the D4 prevents the server from iterating through every example in $\mathbb{S}$ for responding to every data consumer. Our problem setting differs from that of NDS in that we consider the number of sources $M$ as an extrinsic variable, i.e. something outside the control of the server. This necessitates a solution that scales cheaply with the number of sources, as stated in D4.

## 2.2 Neural Data Server

NDS partitions the source data through either (1) superclass partition or (2) unsupervised partition. In either case, image features are extracted with a pretrained network. The image features are either (1) averaged over samples of the same class and then clustered with k-means, or (2) clustered with k-means directly. We refer to each of these clusters as $S_i$ ($i$ ranges from 1 to $K$) as these are analogous to the sources considered in our problem formulation. Then, expert networks are trained for rotation prediction on each partition. Specifically, for each image $\mathbf{x}$ and a rotation angle $\theta \in \{0, 90, 180, 270\}$, let $r(\mathbf{x}, \theta)$ denote the image rotated clockwise by $\theta$. An expert is a function $f : \mathcal{X} \to \Delta^3$, where $\Delta^n$ is a probability mass function of $n + 1$ choices. The loss minimized by a expert $f_i$ is: $\mathcal{L} = -\sum_{\mathbf{x} \in S_i} \sum_{j=0}^{3} \log f_i(r(\mathbf{x}, j))_j$.

Once an expert is trained, accuracy of image rotation prediction is evaluated on the target set:

$$z_i = \frac{1}{4|S_i|} \sum_{\mathbf{x} \in T} \sum_{j=0}^{3} \mathbb{1}(\arg \max_k [f_i(r(\mathbf{x}, j))_k] == j). \tag{2}$$

The importance weight $w_i$ for each partition $i$ is computed as *softmax*$(\mathbf{z})_i$. The sample weight for each example in the source is computed as $\pi(\mathbf{x}) = \sum_{i}^{K} \mathbb{1}(\mathbf{x} \in S_i) \frac{w_i}{|S_i|}$, which is used to sample examples without replacement from the union of source data sets. Exposing the models directly trained on the source datasets in NDS to the client introduces privacy leakage, as the trained experts may memorize certain examples in the source data. It has been shown that is possible to reconstruct training images from trained CNNs by optimizing images from noisy priors [19, 47, 11]. We note that while NDS satisfies D1, it does not satisfy D4 and there may also be privacy leak on D2.

## 3 Scalable Neural Data Server

Scalable Neural Data Server (SNDS) is a data search engine for transfer learning. It is designed with application to large data sharing platforms in mind, where each data provider holds a source dataset. The server's goal is to find the most relevant data to a data consumer who has a budget constraint for the amount of data they could afford. Since it is not feasible to enumerate through every example indexed by the server when responding to every consumer, we simplify the search problem from "find the most relevant examples" to "find the most relevant source datasets". This simplification was also implied by NDS, who uses experts to represent examples of each source split.

SNDS is scalable in that the number of experts downloaded by each data consumer, as well as the amount of compute used in the transaction, remains constant regardless of the number of data providers. This satisfies the computational cost constraint, making it scalable to a large number of source datasets. SNDS is also privacy preserving for both the data consumer and the data providers: no source or target data, nor any models trained on either data, is moved until the recommendation is made. We introduce an intermediary public dataset for training a fixed set of experts. Outputs of these experts on source and target data are used as feature for matching sources to targets.

### 3.1 SNDS Overview

In SNDS, we assume that the server has access to a public dataset $P$. The server partitions $P$ into $K$ disjoint subsets, which we denote as $\{P_k\}_{k=1}^{K}$. The server then optimizes an expert $E$, belonging to

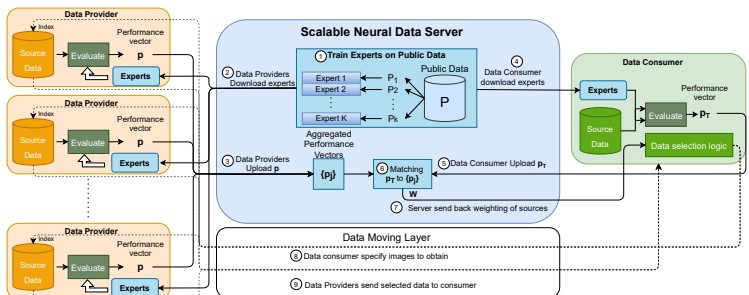

Figure 2: Diagram of our Scalable Neural Data Server for recommending data to a single consumer. Step 1 occurs during initialization. Steps 2 & 3 occur during indexing of data providers. Steps 4-7 occur during data selection. Steps 8-9 are data transactions external to SNDS.

some hypothesis class $\mathcal{E}$, for some objective $L$ with learning algorithm $\mathcal{A}_L$, for each $P_k$. We thereby obtain $K$ experts $\{E_k\}_{k=1}^K$. The server defines a performance function $\mathcal{P}(D, E) : \mathcal{X}^n \times \mathcal{E} \to \mathbb{R}$, which computes some metric using $E$ on a dataset $D$. The output of $\mathcal{P}$ is a single real number, but can also be vector valued. Each data provider $i$ runs the performance function on their data $S_i$. This provides a $K$ dimensional representation $\mathbf{p}_i = (p_{i,1}, \ldots, p_{i,K})$ for each $S_i$ in $\mathbb{S}$, where $p_{i,k} = \mathcal{P}(S_i, E_k)$. At the same time, a data consumer also evaluates the experts on their target dataset $T$ with $\mathcal{P}$ to obtain a representation $\mathbf{p}_T = (p_{1,T}, \ldots, p_{K,T})$. These representations are communicated back to the server, who then computes similarity between the target representation and that of each source. Let $sim(a, b) : \mathbb{R}^K \times \mathbb{R}^K \to \mathbb{R}$ denote a similarity function. We then have $\mathbf{z} = (z_1, \ldots, z_M | z_i = sim(\mathbf{p}_i, \mathbf{p}_T))$ as the similarity scores between the target and each source. Similarity scores are provided to the data consumer as the server's recommendation of sources. The data consumer may then use the similarity scores to select data sources for transfer learning.

### 3.2 Server Initialization

In our experiments, we use the ImageNet Large Scale Visual Recognition Challenge (ILSVRC) 2012 training set as the public dataset. $K$-means on image features is used to partition the ILSVRC2012 training split into 50 partitions. Our main requirement for the experts is that their output as observed by $\mathcal{P}$ correlates with transfer performance from their partition of the public dataset. Following NDS, we use image rotation prediction as the task for training experts, as this task has been shown to correlate well with domain confusion, which is in turn an empirical estimate of transfer performance. As such, we train $E_k$ with loss function $L(P_k, E_k)$, where $L(D, E) = - \sum_{\mathbf{x} \in D} \sum_{l=0}^3 \log E(r(\mathbf{x}, l))_l$. Since experts only need to be trained once during initialization, the computational cost of this step does not contribute to the marginal cost of indexing or queries.

### 3.3 Indexing Sources

To add a source $S_i$ to SNDS, we evaluate experts $\{E_k\}_{k=1}^K$ on $S_i$ and store the results $\mathbf{p}_i$ on the server. First, the experts trained in section 3.2 are downloaded by the data providers. Then, since the experts are trained for image rotation prediction, we use the rotation prediction accuracy as the evaluation metric. Specifically, $\mathcal{P}(D, E) = \frac{1}{4|D|} \sum_{\mathbf{x} \in D} \sum_{j=0}^3 \mathbb{1}(\arg\max_o [E(r(\mathbf{x}, j))_o] == j)$. The representation of a source $S_i$ in the server is thus $\mathbf{p}_i = (\mathcal{P}(S_i, E_k))_{k=1}^K$. The computational cost of this step for data provider $i$ is $\propto K m_i$, which satisfies the complexity constraint of $O(m)$.

### 3.4 Data Selection

A data consumer queries SNDS by first downloading experts $\{E_k\}_{k=1}^K$ and then evaluating each expert on their target dataset $T$. Similar to Sec. 3.3, evaluation function $\mathcal{P}$ is used to obtain $\mathbf{p}_T = (\mathcal{P}(T, E_k))_{k=1}^K$, which is communicated back to the server. The asymptotic computational complexity of producing $\mathbf{p}_T$ is $O(n)$, which meets our requirements listed in Sec. 2. Furthermore, the number of downloaded experts and evaluations is independent of $M$ sources, and so SNDS satisfies D4 on the consumer-side.

The server needs to weigh each source. For simplicity and computational efficiency, we choose a normalized variant of cosine similarity because it is shift invariant in its input arguments and has a fixed output range. This alleviates the need to tune the selection strategy based on the performance metric. Specifically, we compute the channel mean as $\bar{\mathbf{p}} = \frac{1}{M} \sum_{i=1}^M \mathbf{p}_i$, and use this to obtain centered performance vectors $\mathbf{p}'_i = \mathbf{p}_i - \bar{\mathbf{p}}$ and $\mathbf{p}'_T = \mathbf{p}_T - \bar{\mathbf{p}}$. We define the similarity function as $sim(\mathrm{a}, \mathrm{b}) = \frac{||\mathrm{a} \cdot \mathrm{b}||^2}{||\mathrm{a}|| ||\mathrm{b}||}$, so that $\mathbf{z} = (\frac{||\mathbf{p}'_i \cdot \mathbf{p}'_T||^2}{||\mathbf{p}'_i|| ||\mathbf{p}'_T||})_{i=1}^M$.

We normalize $\mathbf{z}$ via softmax with temperature $\tau$ to obtain source weights $\mathbf{w} \in \mathbb{R}^M$. The temperature parameter determines the trade-off between sampling greedily from the most relevant clusters and sampling uniformly from all clusters. Rather than assuming a constant value for $\tau$, we find the $\tau$ that satisfies a target entropy value for $\mathbf{w}$. Intuitively, this allows us to maintain the same trade-off between greedy vs. uniform sampling across different consumer datasets, and this approach outperforms a fixed $\tau$ in practice. More details on this in the supplementary material.

Finally, the server outputs $\mathbf{w}$ to the data consumer, who uses it to select source data. Computing $\mathbf{w}$ from the expert scores incurs an asymptotic cost of $O(MK)$ to the server. This is the main operation that scales with the number of data providers in the system. We note that millions of dot products between low (K) dimensional vectors can be performed in seconds on modern hardware, and the computational cost in both NDS/SNDS is vastly overshadowed by the expert model evaluations on the consumer's data – which is precisely the cost SNDS is trying to minimize.

### 3.5 Differences with NDS

SNDS implements many similar techniques as NDS, but has significant differences to NDS conceptually. We highlight the main differences between SNDS and NDS:

1. NDS is designed to find data from few datasets. SNDS is designed to find data from many datasets.

2. NDS trains an expert for each source. SNDS trains an expert for each intermediary public dataset.

3. NDS uses performance of experts on the target dataset as proxy for similarity to sources. SNDS measures performance of (public) experts on sources and targets, and then uses similarity in performances as proxy for data similarity.

4. Querying with a target dataset of size $n$ on NDS incurs a bandwidth cost $\propto M$ and computational cost $\propto M \times n \times C$. Querying with a target dataset of size $n$ on SNDS incurs a bandwidth cost $\propto K$ and computational cost $\propto K \times n \times C$. Here $C$ is the cost of evaluating an expert on a data point.

### 3.6 Theoretical Analysis

The benefits of SNDS are achieved by disentangling the training of experts from sources: the computational and privacy drawbacks are due to the 1-to-1 correspondence between sources and experts. This "transfer by proxy" approach works because intuitively, if tasks A and B are both "similar" to C, then A and B are also "similar" to each other. Indeed, theory in domain adaptation supports this intuition [8, 28, 4]. For example, Mansour et al. [28] have provided a generalization bound on the target domain.

**Theorem 3.1.** *Mansour et al. [28]: Let $S$ and $T$ be the source and target domains over $\mathcal{X} \times \mathcal{Y}$, $\mathcal{H}$ be a hypothesis class, and let $l : \mathcal{Y} \times \mathcal{Y} \to \mathbb{R}^+$ be a symmetric loss function that obeys the triangle inequality. Further, let $h_S^* = \mathrm{argmin}_{h \in \mathcal{H}} R_S^l(h)$ and $h_T^* = \mathrm{argmin}_{h \in \mathcal{H}} R_T^l(h)$ denote the optimal hypothesis for their respective domains, we have*

$$\forall h \in \mathcal{H}, R_T^l(h) \leq R_S^l(h, h_S^*) + disc_l(S_\mathcal{X}, T_\mathcal{X}) + R_T^l(h_T^*) + R_S^l(h_T^*, h_S^*),$$

*where $R_S^l(h, h_S^*) = E_{\mathbf{x} \sim S}[l(h(\mathbf{x}), h_S^*(\mathbf{x}))]$; $S_\mathcal{X}$ and $T_\mathcal{X}$ are source and target distributions marginalized over $\mathcal{X}$.*

We make the simplifying assumption that the source task is the same as the target task, since it is not computationally feasible for the server to identify the task of every source and target. By the same vein, we assume that the risk on any $S$ is similar. Thus the main quantity we look to optimize is $disc_l(S_\mathcal{X}, T_\mathcal{X})$. Which is defined as:

$$disc_l(S, T) = \sup_{(h, h') \in \mathcal{H} \times \mathcal{H}} \left| R_S^l(h', h) - R_T^l(h', h) \right| \tag{3}$$

An important property of $disc_l$ is that it obeys triangle inequality independently of the choice of $l$. That is: $disc_l(S, T) \leq disc_l(S, P) + disc_l(P, T)$ for some intermediate domain $P$ over the same subspace $\mathcal{X}$. A simple observation is then that if we minimize the upper bound, we are indeed minimizing the $disc(S, T)$. This observation fuels the motivation for our approach.

# 4 Experiments

We first empirically verify that our similarity function output is well correlated with domain confusion, validating our choices of the proxy task and similarity function in the data selection process. We then experimentally evaluate SNDS in the task of classification. We use accuracy of classifier trained through transfer learning on target datasets to gauge the usefulness of data recommended by SNDS. We also extend the experiments into domains dissimilar to ImageNet, to test the generality of SNDS.

## 4.1 Datasets

**Source data** We simulate data providers with partitions of the OpenImages dataset[26]. The Open-Images dataset contains 9 million images with various modes of annotations. We split OpenImages into 50 partitions using $K$-means on image features extracted by an ImageNet pretrained network. Each image in OpenImages is tagged with multiple object bounding boxes with labels. We convert OpenImages into a classification dataset by cropping the images by the bounding boxes and assigning the label of that bounding box as label of the cropped image. Thus, we obtain a classification dataset of 15.9 million cropped images, each with a label from the 600 object bounding box categories.

**Public data and expert training** We use the training split of ILSVRC2012[37], containing 1.2 million images, as the public dataset. We split it using superclass partition as described in [45]: the mean features of each class are extracted using an ImageNet pretrained network, and then clustered using $K$-means. Experts are trained on the task of image rotation prediction. Experts use ResNet18 as backbone with a input size of 224x224 and output dimension of 4 (corresponding to the 4 rotations). We use $K = 50$ in our experiments, and study the effect of reducing $K$ in an ablation. Further details on transfer learning are found in the appendix. Experiments are implemented using PyTorch[34].

**Target Datasets** We use nine finegrained classification datasets as target datasets. They are: FGVC-Aircraft [27], Stanford Cars [25], CUB200 [43], Stanford Dogs [24], DTD [13], Flowers102 [31], Food100 [9], Oxford Pets [33], and SUN397 [44].

## 4.2 Domain confusion and SNDS

Since we want SNDS to select sources that are similar to the target domain, the scores of each source returned by SNDS should intuitively be correlated with $disc_l(S_i, T)$ for source $S_i$ and target $T$. Domain confusion [8, 21] has been used as an empirical approximation of $disc_l(A, C)$. In this experiment, we compare domain confusion and score provided by SNDS for 3 target datasets.

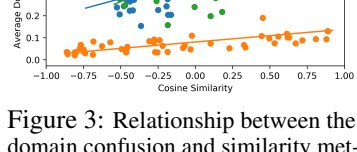

Figure 3: Relationship between the domain confusion and similarity metric of the source and target domains.

We use an MLP with 1 hidden layer of 512 neurons to classify between each pair of source and target. We use a weighted sample of 50k images from the mixed source/target dataset to ensure that the samples are balanced, as done in NDS. We report the test loss on a held-out set of images from the source and target domains.

As seen in Figure 3, the score returned by SNDS is positively correlated with the domain confusion between the source and target domains. This shows that SNDS indeed outputs higher scores for sources similar to the target domain. This can also be observed qualitatively in Figure 4.

## 4.3 Data Recommendation with SNDS

**Implementing Data Consumers** Evaluation of SNDS on downstream tasks requires us to experimentally simulate data consumers. For target domain data, we use the finegrained classification tasks described earlier - each classification task represents one data consumer. Once a recommendation of sources has been made by the server, it is up to the data consumer to decide on their budget, data selection strategy, and transfer learning method. Similar to NDS, our simulated data consumer sample from all available data in $\mathbb{S}$ without replacement, weighing the likelihood of each data point as $\pi(\mathbf{x}) = \sum_i^M \mathbb{1}(\mathbf{x} \in S_i)\frac{\mathbf{w}_i}{|S_i|}$. We pre-train on the selected data using supervised learning, and then finetune on the downstream dataset. Details about our transfer learning procedure are in the appendix. Experiments are performed on a Tesla P100 with 12 GB memory in an internal cluster.

Table 1: Downstream task performance with pretraining data recommended by SNDS.

| Selection Method | % Images | Target Dataset | | | | | | | | | Average |
|---|---|---|---|---|---|---|---|---|---|---|---|
| | | CUB200 | Flowers102 | Pets | Food 101 | Stanford Dogs | DTD | Cars | Aircraft | SUN397 | |
| No pretraining | 0% | 27.25 | 52.42 | 42.21 | 71.75 | 39.35 | 30.16 | 18.78 | 45.80 | 31.80 | 39.95 |
| Random Sample | 2% (292K) | 41.49 | 74.03 | 67.02 | 72.34 | 52.93 | 52.72 | 46.77 | 54.66 | 31.80 | 55.75 |
| SNDS-50 | 2% (292K) | 50.59 | 81.66 | 71.06 | 73.73 | 55.07 | 56.28 | 47.37 | 55.91 | 39.79 | 59.07 |
| NDS | 2% (292K) | 49.91 | 78.32 | 70.71 | 73.35 | 54.64 | 55.32 | 52.95 | 57.95 | 42.27 | 59.53 |
| Random Sample | 5% (730K) | 53.11 | 83.53 | 73.79 | 74.86 | 57.93 | 57.87 | 67.34 | 62.70 | 44.42 | 63.95 |
| SNDS-50 | 5% (730K) | 57.35 | 87.68 | 75.91 | 74.87 | 59.38 | 60.59 | 63.80 | 62.32 | 43.99 | 64.96 |
| NDS | 5% (730K) | 59.04 | 85.10 | 75.18 | 75.58 | 58.99 | 60.27 | 66.56 | 63.76 | 45.98 | 65.61 |
| Random Sample | 10% (1.46M) | 58.40 | 86.82 | 76.40 | 76.22 | 60.81 | 60.74 | 71.58 | 65.24 | 47.69 | 67.10 |
| SNDS-50 | 10% (1.46M) | 61.02 | 89.89 | 77.32 | 77.29 | 61.33 | 64.41 | 72.68 | 65.12 | 45.89 | 68.33 |
| NDS | 10% (1.46M) | 61.27 | 89.88 | 78.32 | 77.03 | 61.28 | 60.96 | 73.77 | 65.57 | 46.84 | 68.33 |

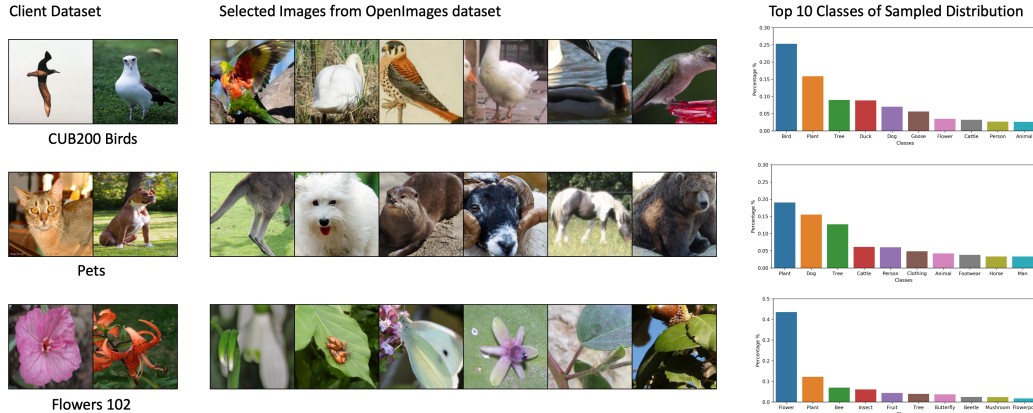

Figure 4: From left to right: Image from client dataset, images from "most similar" public split (ImageNet), selected images from data sources (Openimages).

### 4.3.1 Main Experimental Results

Table 2: Downstream performance on Pets & CUB200 Birds, using pretraining data recommended by SNDS with no target class in public data

| Selection Method | % Images | Target Dataset | |
|---|---|---|---|
| | | CUB200 | Oxford Pets |
| Random Sample | 2% (292K) | 41.49 | 67.02 |
| SNDS | 2% (292K) | 50.59 | 71.06 |
| SNDS (no target data) | 2% (292K) | 47.67 | 70.35 |
| Random Sample | 5% (730K) | 53.11 | 73.79 |
| SNDS | 5% (730K) | 57.35 | 75.91 |
| SNDS (no target data) | 5% (730K) | 58.69 | 76.83 |

In this experiment, we select 2% (292K images), 5% (730K images) and 10% (1.46M images) of total source data using SNDS and compare with randomly selected data and NDS. We implement NDS with image-rotation experts, as per Sec. 2. Our results demonstrate that data selected by SNDS is more useful in almost every downstream task than randomly selected data. This advantage is most pronounced when using a tight budget of 2%, since the chance of randomly selecting useful data is higher when more data is being selected.

Furthermore, we find that SNDS is competitive with NDS even though NDS trains experts directly on source data, whereas SNDS trains experts on the public dataset. As such, SNDS promises cheaper computation cost with growing number of sources without sacrificing much in performance.

Fig. 4 shows retrieval results from SNDS for several target datasets. Similarity between the target and the public data split is determined by the accuracy of that split's expert on the target dataset. Since OpenImages provides multiple class labels per image, the top classes in the sampled distribution may be ones that co-occur with more relevant classes, such as the *Plant* class for the Pets dataset.

We ablate the use of 50 experts in the appendix, and find that with just 10 experts, SNDS is able to out-perform the baseline, though having more experts allows for more specific recommendations that improve the downstream performance.

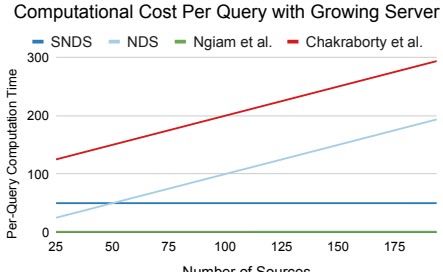
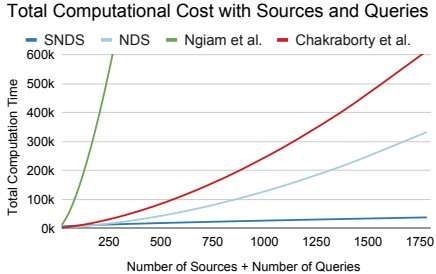

Figure 5: Comparison of simulated computational cost.

**Impact of missing public data**    A critical factor determining the feasibility of SNDS is whether it generalize to types of images that are not represented in the public dataset. That is, it should be able to recommend source images similar to those in the target even if no such images are in the public dataset. We emulate this by dropping images in the public dataset with classes corresponding to the target task. Two target tasks: Oxford Pets and CUB200, are used. After we split the public dataset into 50 superclass clusters, we remove clusters that contain overlap with target domain classes. For the Oxford Pets experiment, we removed eight clusters that included images containing dogs and cats, and for the CUB200 experiment, we removed nine clusters that contained images of birds. The remaining clusters are used to train experts and in turn recommend data. From Table 2, we can see that dropping experts trained on classes of images in the target set did not have significant effect on performance. Hence, SNDS can be used in the real world without frequent updates to the public dataset once it is deployed. As a side note, we use sampling without replacement over the entire source dataset, similar to that used in NDS, for SNDS results reported in this experiment.

**Simulating Computational Cost**    We simulate the computational cost of SNDS and compare it to that of Yan et al. [45], Ngiam et al. [30], and Cui et al. [15] in Fig. 5. We assume that the number of sources and the number of queries both grow over time, while the number of images per source stays constant. Detailed modeling assumptions can be found in the supplementary material. For the data consumer, SNDS has a flat per-query computational cost that does not grow with additional sources. Over time lifetime of a server, SNDS's total computational cost (incurred by all parties) is also the lowest among the methods compared.

### 4.3.2    Scaling beyond the Natural Image Setting

The computational benefits of SNDS stem from using a fixed set of experts for performing data recommendation. For the system to scale, these experts must be able to provide clients with recommendations in domains beyond the ones that they has been trained on. To test this, we explore the performance of the server, both qualitatively and quantitatively, when recommending data for domains that are highly dissimilar to natural images and find that SNDS is able to provide useful recommendations using experts that have been trained on ImageNet. Note that in these experiments the public experts are kept the same, i.e. experts trained on ImageNet.

**Medical data**    Here, we focus on whether SNDS is able to recommend meaningful medical data with experts trained only on ImageNet. We add the chest x-ray dataset CheXpert [23] and the diabetic retinopathy dataset provided by Kaggle [2, 14] to the original source datasets consisting of OpenImages clusters. We examine the recommendations that the server provides when using different targets datasets of the same modalities as the source datasets: the NIH ChestX-ray14 dataset [42], and the Aptos 2019 diabetic retinopathy dataset [1]. We do not perform any additional pre-processing to these datasets when incorporating them within SNDS.

In Fig. 6, we show the softmax scores of the top 5 recommended clusters for both the NIH ChestX-ray14 and Aptos2019 datasets. In both cases, the most relevant cluster in the source data corresponds to the image type in target data. Furthermore, SNDS does not conflate the recommendations between the chest x-ray and diabetic retinopathy, suggesting that the server is not grouping all dissimilar data together and maintains its ability to provide valuable recommendations in highly dissimilar data.

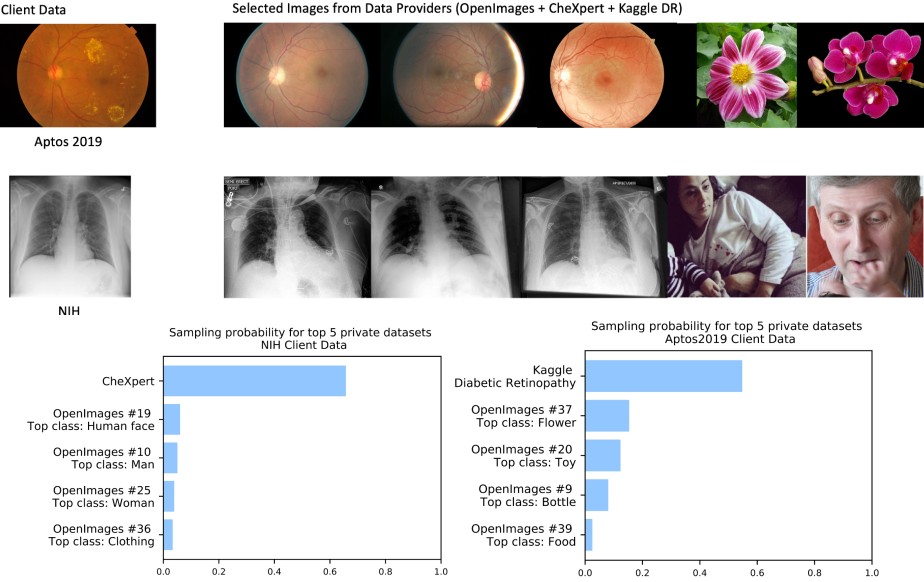

Figure 6: Recommended data and output probabilities for satellite and x-ray data

In practice, the best method for pre-training with medical images remains an open question and may not provide consistent benefits [36], though recent works in self-supervised learning have shown improvements when using a multi-step training process [6]. In our experiments, a naive supervised learning approach did not provide significant transfer benefits, even when directly transferring from one x-ray dataset to another (the oracle recommendation scenario). Still, the relevant recommendations provided by our server are directly usable by the consumer for improving performance using the latest techniques in this area as they emerge.

**Sketch and satellite data evaluation** Due to the challenges associated with transfer learning in the medical setting, we evaluate the benefits of transfer on more balanced datasets consisting of satellite and sketch datasets. We add the Quick-

Table 3: Task Performance for Satellite & Sketch Data

| Selection Method | % Images | Target Dataset | |
|---|---|---|---|
| | | Sketch-Mini | NWPU-RESISC45-Mini |
| Rnd. Sample (+ sketch/sat) | 0.5% (73K) | 11.79 | 52.40 |
| SNDS (OpenImages only) | 0.5% (73K) | 13.60 | 53.60 |
| SNDS (+ sketch/sat data) | 0.5% (73K) | 22.50 | 54.80 |

Draw dataset split from [35] and the PatternNet dataset [48] to the source sets of our server. As the client, we evaluate the performance of our method when using 1k subsets of the Sketch [18] and NWPU-RESISC45 datasets [12], which we denote as Mini versions of those datasets. We use these subsets rather than the full datasets to better observe the effects of pretraining, as was done in NDS. To perform pretraining with the recommended data, we mix the various datasets together and perform standard supervised training with cross-entropy loss, predicting the class of the image from the union of all the dataset classes.

Table 3 provides performance for the satellite and sketch datasets. In both domains, SNDS shows performance benefits over a random sample of data from the server. SNDS even shows benefits over a random sample *without* satellite and sketch dataset in the server, which suggests that our server is able to find natural images that help with downstream performance, even in dissimilar domains. The experts used for recommendation have been trained exclusively on ImageNet clusters, demonstrating that SNDS is truly able to scale and improve task performance in highly dissimilar domains.

We see transfer benefits using our simple pre-training approach, which leads us to believe that consumers will be able to benefit even more from the data recommendations when using more complex approaches from multi-task learning. This highlights the strength and robustness of the SNDS' recommendations, which provides immediate benefits with a simple transfer algorithm.

**Limitations** While SNDS can be applied to many task types, our experiments have focused on fine-grained classification tasks, allowing us to illustrate its behaviours across different data domains

and in several important ablations. We also omit transfer experiments in the medical domain, as mentioned in Sec. 4.3.2.

# 5 Related Work

Our work is built upon the rich literature of transfer learning and domain adaptation [49, 32, 8, 4, 28]. Of most resemblance to ours are the "instance weighing" approaches [39, 40, 7, 16, 15, 30, 3, 10]. These methods compute the weight of instances or sets of instances based on some measure of usefulness to the target task. Our work differs from these methods in data separation and computational constraints: our problem setting requires a method that does can operate with the source and target dataset on separate devices, and the computational cost for indexing new source data must be low.

Cui et al. [15] uses similarity between mean features of each class as extracted by a NN trained on source data to match classes of the source to target dataset. Ngiam et al. [30] instead labels the target dataset with a NN trained on source data, and recommends source classes that appear most frequently as pseudo-labels. These methods requires retraining of the NN whenever the source data is updated, thus they are computationally expensive in terms of total cost.

Achille et al. [3] find the Fisher information matrix for each source and target dataset to compute a similarity measure between sources and targets. This approach requires access to the consumer's labels and is challenging to scale beyond classification tasks, and thus not immediately applicable to our setting where we focus on being label-agnostic to allow scalability to arbitrary tasks.

Chakraborty et al. [10] proposed to train a NN on the target dataset, and compute similarity between the target dataset and each source in the feature space of this NN. Finally, Yan et al. [45] proposed NDS, reviewed in Sec. 2. These two methods have a per-query computational cost that grows with the number of sources. Furthermore, all existing methods require training of NNs on either sources or targets. This violates our privacy constraints (D1 and D2 in desiderata). SNDS protects the privacy of source and target data, and has a lower computational cost than existing methods.

# 6 Conclusion

We proposed SNDS, a scalable data recommendation method that selects sets of useful pretraining data for each given downstream task. In SNDS, sources and targets are represented by their degree of similarity to public datasets. The biggest advantage of SNDS over existing methods is computational efficiency: the computational cost for data consumers remains constant with the number of sources indexed by the server, and sources already indexed by SNDS do not need to be re-indexed when adding new sources. Experiments show that SNDS can work well even if there is data in the public datasets that is dissimilar to target, thereby allowing SNDS to generalize beyond what is contained in the public datasets. Lastly, we demonstrated the robustness of SNDS by applying it to a variety of domains dissimilar to natural images and show transfer benefits in that setting.

**Potential Societal Impact**   SNDS is a large-scale search engine for transfer learning data. Users can use SNDS to find relevant pretraining data for their dataset, which allows practitioners with smaller datasets to leverage the latest models designed for larger datasets, and to select pretraining data according to their computational needs. This is especially beneficial for consumers for whom collecting a large dataset is cost-prohibitive, such as in hospitals or non-profit organizations. As a data recommendation search engine, our system is susceptible to misuses. In a completely open system, data providers may upload inappropriate datasets or biased data to intentionally or unintentionally cause harm to the consumer. This can be mitigated by having a dataset approval process for SNDS and by including the latest tools to explore dataset biases as part of the server.

**Acknowledgment**   This work was partially supported by LG and NSERC. SF acknowledges the Canada CIFAR AI Chair award at the Vector Institute. TC acknowledges support from Vector Scholarship for Artificial Intelligence. DA acknowledges support from the Microsoft Ada Lovelace Fellowship.

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
