## A  Transfer Learning Details

Experiments are performed in PyTorch, with licensing information here: `https://github.com/pytorch/pytorch/blob/master/LICENSE`. Pretraining is performed using a ResNet18 as backbone on the selected data. We train for 40 epochs on the classification task with cross-entropy loss, and then finetune for 100 epochs on the target dataset with a new classification head. In Section 4.3.1 the task is classification between the 601 classes in OpenImages, whereas in Sec4.3.2 we classify between all of the classes in the mixed data recommendations. We use performance on validation split of the target dataset for early stopping during finetuning. No early stopping is used during pretraining. SGD optimizer with a momentum $0.9$ is used for both pretraining and finetuning, with learning rates of $0.1$ and $0.01$ respectively. We decay the learning rate of the optimizer during finetuning by a factor of 10 every 30 epochs. Weight decay of $0.0001$ is applied during both steps. All images are normalized in each channel by Imagenet image normalization weights. For data augmentation during pretraining, we resize each image to a short-edge length of 256 and then randomly crop a 224x224 region from the image. Random horizontal flipping is also used. During finetuning, we additionally perform random rotation of up to 20 degrees.

## B  Proof of triangle inequality

Recall notation from the main text: $R_A(h, h')$ is the risk associated with predicting with hypothesis $h$ on data distribution $A$ when the true labeling function is $h'$. We define $disc_l(S, T)$ as:

$$disc_l(S, T) = \sup_{(h,h') \in \mathcal{H} \times \mathcal{H}} \left| R_S^l(h', h) - R_T^l(h', h) \right| \tag{4}$$

Let $(h_{AC}, h'_{AC})$ denote the pair of hypothesis that maximizes $\left| R_A^l(h', h) - R_C^l(h', h) \right|$, then

$$disc_l(A, C) = \left| R_A^l(h_{AC}, h'_{AC}) - R_C^l(h'_{AC}, h_{AC}) \right| \tag{5}$$

$$\leq \left| R_A^l(h_{AC}, h'_{AC}) - R_B^l(h'_{AC}, h_{AC}) \right| \tag{6}$$

$$+ \left| R_B^l(h_{AC}, h'_{AC}) - R_C^l(h'_{AC}, h_{AC}) \right| \tag{7}$$

$$\leq \sup_{(h,h') \in \mathcal{H} \times \mathcal{H}} \left| R_A^l(h, h') - R_B^l(h', h) \right| \tag{8}$$

$$+ \sup_{(h,h') \in \mathcal{H} \times \mathcal{H}} \left| R_B^l(h', h) - R_C^l(h', h) \right| \tag{9}$$

$$= disc_l(A, B) + disc_l(B, C). \tag{10}$$

## C  Description of Datasets

The datasets used in our experiments are described in Table 4. We report the appropriate metric (top-1 accuracy or mean-per-class accuracy) as outlined in the evaluation protocols for each dataset. The DTD and SUN397 dataset include multiple train/test splits and we report our results on the first split, as done in prior work. For the CheXpert dataset, we use the splits defined by [29], which creates a larger validation set than the one in the original work. For the mini datasets, we use two separate 1k subsets for training and testing, chosen from the original training datasets while maintaining the class balance.

OpenImages is released under the Apache License 2.0. Imagenet is released for research-purposes only, detailed at `https://image-net.org/download.php`. FCVC-Aircraft is released "for non-commercial research purposes only". Stanfard Cars uses the same license as Imagenet. CUB200 Birds is provided by [43] on their project website `http://www.vision.caltech.edu/visipedia/CUB-200.html`. Stanford Dogs dataset is derived from Imagenet, and made available at `http://vision.stanford.edu/aditya86/ImageNetDogs/`. Describable Textures Dataset is released for research purpose only. Flowers 102 is licensed under the GNU General Public License, version 2. Food-101 was curated by [9] under fair use of data from foodsplotting.com, which has ceased operation in 2018. Oxford-IIIT Pets is available under the Creative Commons Attribution-ShareAlike 4.0 International (CC-BY-SA 4.0). SUN397 is made available for academic research by authors of [44] on `https://vision.princeton.edu/projects/2010/SUN/`. The

| Dataset | Images | Class Count | Evaluation Metric |
|---|---|---|---|
| OpenImages [26] | 1,743,042(images)/ 14,610,229(boxes) | 601 | - |
| Imagenet [17] | 1,281,167 | 1000 | - |
| FGVC-Aircraft [27] | 3334(train) / 3333(val) | 102 | Mean Per-Class |
| Stanford Cars [25] | 8144(train) / 8041 (val) | 196 | Top-1 |
| CUB200 Birds [43] | 5994(train) / 5794(val) | 200 | Top-1 |
| Stanford Dogs [24] | 12,000(train) / 8580(val) | 120 | Top-1 |
| Describable Textures Dataset [13] | 1880(train) / 1880(val) | 47 | Top-1 |
| Flowers 102 [31] | 2040(train) / 6149(val) | 102 | Mean Per-Class |
| Food-101 [9] | 75,750(train) / 25,250(val) | 101 | Top-1 |
| Oxford-IIIT Pets [33] | 3680(train) / 3369(val) | 37 | Mean Per-Class |
| SUN397 [44] | 76,128(train) / 10,875(val) | 397 | Top-1 |
| Kaggle Diabetic Retinopathy [2, 14] | 35,126 (train) / 53,576(test) | 5 | - |
| Aptos 2019 [1] | 3662(train) / 1928 (test) | 5 | - |
| CheXpert [23, 29] | 50,000 (train) / 50,000 (test) | 5 | - |
| NIH ChestX-ray14 dataset [42] | 86,524 (train)/25,596(test) | 14 | - |
| PatternNet [48] | 30,400(train) | 38 | Top-1 |
| NWPU-RESISC45 [12] | 31,500(train) | 45 | Top-1 |
| QuickDraw [35] | 120,750(train)/51,750(test) | 345 | Top-1 |
| Sketch [18] | 20,000 (train) | 250 | Top-1 |

Table 4: An overview of the public, private and target datasets used in our experiments.

Kaggle Diabetic Retinopathy dataset is released by [14] for research purposes, as mentioned by the Kaggle organizers here: `https://www.kaggle.com/c/diabetic-retinopathy-detection/discussion/141968`. The Aptos2019 dataset is released by the Kaggle licensing agreement, which is permissable for "non-commercial purposes only" and "academic research" as seen in `https://www.kaggle.com/c/aptos2019-blindness-detection/discussion/107520`. The CheXpert dataset is available for "personal, non-commercial research purposes only", as seen on `https://stanfordmlgroup.github.io/competitions/chexpert/`. Note that this dataset has 5 classes used for evaluation purposes, but 14 are available in total. The NIH ChestX-ray14 dataset can be used in an "unrestricted" fashion according to the dataset creators as mentioned in `https://nihcc.app.box.com/v/ChestXray-NIHCC/file/249502714403`. The PatternNet dataset is provided for research purposes and must be cited by the users: `https://sites.google.com/view/zhouwx/dataset`. The NWPU-RESISC45 dataset is said to be "publicly available for research purposes", as per [12]. We use the QuickDraw split available from [35], who include a fair use notice allowing others to use it for academic research: `http://ai.bu.edu/M3SDA/`.The Sketch dataset is released under a Creative Commons Attribution 4.0 International License on their site: `http://cybertron.cg.tu-berlin.de/eitz/projects/classifysketch/`.

The datasets we use were provided with consent by their original curators and do not contain personally identifiable information to our knowledge, and we do not introduce any data curation in our work.

## D   Computational Efficiency Simulation

We compare the computational cost of different data recommendation methods through simulation. The goal of the simulation is to show how each method's cost grows with increasing number of sources indexed by the server and with increasing number of targets that the server need to produce recommendations for.

**Modelling assumptions**   We assume that methods which requires neural network training will train for 100 epochs (e.g. SNDS trains experts on the public dataset, NDS trains experts on sources), each source and target have a size of 10000 units, and passing a unit through a neural network has an "iteration time" of $10^{-4}$ seconds (passing one network over one source has a cost of 1). We make the simplifying assumption that both the inference time and training step time can be approximated by "iter time" for ease of comparison. We also assume that the time for matching two representations (e.g. computing a cosine distance in SNDS or [30]) is $1e-4$. For SNDS, we assume that there are 50 splits of public data ($K = 50$), each split also has 10000 units. For [30], we assume that instead of averaging over source labels, they instead train a domain classifier that distinguishes between sources, such that the mode of the prediction of this classifier on the target dataset indicates the most similar source set.

We consider the computational costs with $M$ data providers, target datasets of size $n$ and source datasets of size $m$. For SNDS, we assume that we have $K$ public splits of data and corresponding experts, as mentioned previously.

For computational cost per query, we use the following formulas:

- [30]: $n \times$ iter time $+ M \times$ match time
- [10]: $n \times$ iter time $\times$ Num epoch $+ M \times m \times$ iter time $+ M \times$ match time
- [45]: $M \times n \times$ iter time $+ M \times$ match time
- SNDS: $K \times n \times$ iter time $+ K \times M \times$ match time

Note that for each data recommendation method, the cost of the term containing $n \times$ iter time dominates the term containing $M \times$ match time. This is because $n \times$ iter time corresponds to a eval or train step with a neural network over the target dataset, while $M \times$ match time corresponds to the time to search among the source providers to convert a computed metric into a sampling probability. This match step is performed with very low dimensional vectors compared to the size of the target dataset (at most K dimensions in the case of SNDS). Hence, compared to [45], SNDS' performance benefits stem from reducing the number of forward passes that the consumer is required to compute, with the match time term being negligible. This is seen graphically in Figure 5.

For total computational cost, we assume that the number of queries grows $10\%$ faster than the number of sources grows. Our simulation starts at 10 sources and 10 queries, and grows to 320 sources and 1462 queries. The total computational cost is approximated by integrating the per-query cost over the growth in queries and total indexing cost of $M$ sources, and then summed together.

For the total indexing cost, we use the following formulas:

- [30]: $m \times$ iter time $\times$ Num epoch $\times M \times (M-1)$
- [45]: $M \times m \times$ iter time $\times$ Num epoch
- SNDS: $M \times m \times$ iter time $\times K$
- [10] does not have a separate indexing cost since it trains a network for each query and then performs indexing; all computational costs are captured in the per-query cost.

## E   Softmax Temperature Optimization

Once we have computed similarity scores $\mathbf{z}$ for each data source, we obtain the source weights with a softmax to find the source weights $\mathbf{w} \in \mathbb{R}^M$. However, we empirically find that the temperature $\tau$ is an important hyperparameter to set within the softmax equation. Mathematically, as $\tau \to \infty$, the softmax output resembles a uniform distribution, where each data source has an equal probability. As $\tau \to 0^+$, the softmax output approaches an $\mathrm{argmax}$ output, assigning 1 to the closest data source and 0 probability to the others. Hence, our procedure to determine a softmax temperature defines a trade-off between sampling uniformly and a greedy $\mathrm{argmax}$ output.

To make this trade-off automatically, we constrain the output distribution of weights $\mathbf{w}$ to have a specific entropy value $H(\mathbf{w})$ *across* datasets. Entropy allows us to measure how "closely" the weights $\mathbf{w}$ resemble a uniform distribution, and by setting a target entropy that the weights must satisfy, the output distribution is forced to have the same balance between a uniform and greedy sampling across all datasets. This approach outperforms a fixed $\tau$ in practice. We adjust $\tau$ with gradient descent, which stably converges within dozens of iterations as this is a convex problem. In all experiments, we set the target entropy value to $1.5$ – the first value we tried.

## F   Additional Experiments

**Ablating Sampling Strategy**    Here, we ablate the sampling strategy from the stochastic sampling strategy presented in the paper on the OpenImages source data. Specifically, one could also perform a greedy selection strategy similar to that used in [15] for data recommendations. Once SNDS has returned the score for each source, we start from the source with the highest score and add data to the pretraining set until the budget has been exhausted. Table 5 compares the downstream accuracy of

Table 5: Downstream task performance with different sampling strategies

| Selection Method | % Images | Target Dataset | | | | | | | | | Average |
| | | Aircraft | Cars | CUB200 | Stanford Dogs | DTD | Flowers102 | Food 101 | Pets | SUN397 | |
| --- | --- | --- | --- | --- | --- | --- | --- | --- | --- | --- | --- |
| Random Sample | 2% | 54.66 | 46.77 | 41.49 | 52.93 | 52.72 | 74.03 | 72.34 | 67.02 | 39.79 | 55.75 |
| SNDS | 2% | 55.91 | 47.37 | 50.59 | 55.07 | 56.28 | 81.66 | 73.73 | 71.06 | 40.04 | 59.07 |
| SNDS Greedy Selection | 2% | 53.46 | 43.86 | 50.54 | 55.12 | 54.36 | 79.53 | 72.16 | 69.91 | 38.97 | 57.55 |
| Oracle (NDS) | 2% | 57.95 | 52.95 | 49.91 | 54.64 | 55.32 | 78.32 | 73.35 | 70.71 | 42.27 | 59.53 |
| Random Sample | 5% | 62.70 | 67.34 | 53.11 | 57.93 | 57.87 | 83.53 | 74.86 | 73.79 | 44.42 | 63.95 |
| SNDS | 5% | 62.32 | 63.80 | 57.35 | 59.38 | 60.59 | 87.68 | 74.87 | 75.91 | 43.99 | 64.96 |
| SNDS Greedy Selection | 5% | 62.11 | 63.45 | 56.58 | 57.02 | 60.64 | 86.91 | 75.26 | 74.72 | 43.34 | 64.45 |
| Oracle (NDS) | 5% | 63.76 | 66.56 | 59.04 | 58.99 | 60.27 | 85.10 | 75.58 | 75.18 | 45.98 | 65.61 |

greedy selection to the stochastic sampling strategy. We focus this ablation on 2% and 5% budgets due to the long pre-training time when using 10%. We find that greedy selection tends to perform better than random sampling, but worse than stochastic selection. Upon further inspection, we find that performance on the pretraining task is significantly lower when using greedy selection than that achieved when using stochastic sampling (25.02% vs 29.44% on 2% samples, 31.61% vs 34.40% on 5% samples). This suggests that while the sources with the highest scores may be most similar to the target, they lack the class diversity needed to make a good pretraining set.

Table 6: Downstream task performance with pretraining data recommended by SNDS-10

| Selection Method | % Images | Target Dataset | | | | | | | | | Average |
| | | Aircraft | Cars | CUB200 | Stanford Dogs | DTD | Flowers102 | Food 101 | Pets | SUN397 | |
| --- | --- | --- | --- | --- | --- | --- | --- | --- | --- | --- | --- |
| Random Sample | 2% | 54.66 | 46.77 | 41.49 | 52.93 | 52.72 | 74.03 | 72.34 | 67.02 | 39.79 | 55.75 |
| SNDS | 2% | 55.91 | 47.37 | 50.59 | 55.07 | 56.28 | 81.66 | 73.73 | 71.06 | 40.04 | 59.07 |
| SNDS-10 | 2% | 57.55 | 50.86 | 43.92 | 53.21 | 55.00 | 81.64 | 71.97 | 69.92 | 40.82 | 58.32 |
| Random Sample | 5% | 62.70 | 67.34 | 53.11 | 57.93 | 57.87 | 83.53 | 74.86 | 73.79 | 44.42 | 63.95 |
| SNDS | 5% | 62.32 | 63.80 | 57.35 | 59.38 | 60.59 | 87.68 | 74.87 | 75.91 | 43.99 | 64.96 |
| SNDS-10 | 5% | 62.19 | 64.03 | 53.09 | 57.91 | 59.68 | 89.29 | 75.91 | 74.65 | 43.28 | 64.00 |

**Ablating Number of Experts** In this ablation we study the effect of reducing the number of experts. We use the same public dataset (ImageNet) as before, but split it 10 ways ($K = 10$) to train 10 experts. We call this smaller dataserver SNDS-10. We proceed to evaluate SNDS-10 on the same target tasks as used in Table 1. Results are reported in Table 6. We find that using fewer experts decreases the performance of SNDS. Our hypothesis is that this is caused by a lack of signal for SNDS to characterize datasets. Specifically, datasets are represented in SNDS by the performance of experts, which is a $K$ dimensional vector with range $[0, 1]$. The capacity of this representation space (i.e. covering number) has an exponential relationship with $K$. When $K$ is reduced from 50 to 10, datasets that had distinct representations are now mapped to similar coordinates, thereby decreasing SNDS's ability to identify useful sources.

Table 7: Top-1 Accuracy on the pre-training task.

| Selection Method | % Images | Target Dataset | | | | | | | | | Average |
| | | Aircraft | Cars | CUB200 | Stanford Dogs | DTD | Flowers102 | Food 101 | Pets | SUN397 | |
| --- | --- | --- | --- | --- | --- | --- | --- | --- | --- | --- | --- |
| Random Sample | 2% | | | | | 34.78 | | | | | |
| SNDS | 2% | 27.98 | 29.99 | 30.84 | 30.77 | 28.86 | 26.22 | 29.97 | 30.85 | 29.44 | 29.44 |
| SNDS Greedy Selection | 2% | 22.21 | 27.07 | 26.12 | 29.48 | 26.54 | 15.81 | 23.61 | 27.91 | 26.41 | 25.02 |
| Random Sample | 5% | | | | | 39.53 | | | | | |
| SNDS | 5% | 32.16 | 33.14 | 36.37 | 35.08 | 34.34 | 34.68 | 35.16 | 34.83 | 33.88 | 34.40 |
| SNDS Greedy Selection | 5% | 29.76 | 32.40 | 30.44 | 31.60 | 34.83 | 29.01 | 30.93 | 31.75 | 33.76 | 31.61 |

**Pretraining task accuracy** We present the accuracy achieved on the 601-way classification pre-training task with OpenImages sources in Table 7. For the "Random Sample" baseline, all downstream tasks share the same pretrained network, hence only one performance number is reported.

**Repeat runs with confidence intervals** We perform repeat trials for the first three target datasets of Table 1 for SNDS and NDS to obtain confidence intervals. In Table 8, we show the average and standard deviation over 3 runs with different seeds for SNDS and NDS. This corresponds to 3 different samples of data for each reported average.

Table 8: Downstream task performance with 3 repeated runs.

| Selection Method | % Images | | Target Dataset | |
| | | CUB200 | Flowers102 | Pets |
| --- | --- | --- | --- | --- |
| SNDS | 2% | $50.24 \pm 0.88$ | $82.66 \pm 0.79$ | $70.94 \pm 1.17$ |
| NDS | 2% | $49.94 \pm 0.28$ | $79.67 \pm 0.80$ | $70.85 \pm 0.57$ |
| SNDS | 5% | $57.67 \pm 1.05$ | $87.02 \pm 0.29$ | $75.42 \pm 0.74$ |
| NDS | 5% | $58.33 \pm 0.61$ | $86.15 \pm 0.13$ | $75.36 \pm 0.57$ |
| SNDS | 10% | $60.81 \pm 0.61$ | $89.94 \pm 0.52$ | $77.09 \pm 0.17$ |
| NDS | 10% | $60.23 \pm 1.59$ | $88.86 \pm 1.77$ | $77.35 \pm 0.96$ |

# G   Additional Qualitative Results

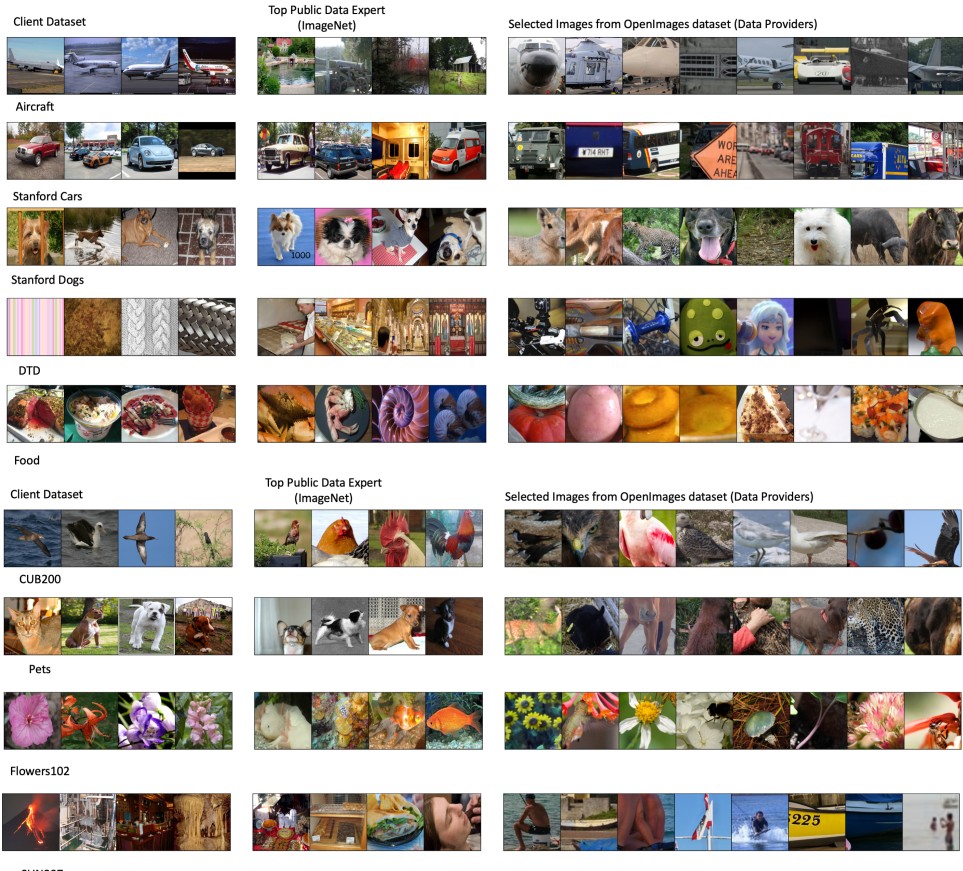

Figure 7: From left to right: Images from client/consumer dataset, images from "most similar" public split (ImageNet), selected images from data sources (Openimages).

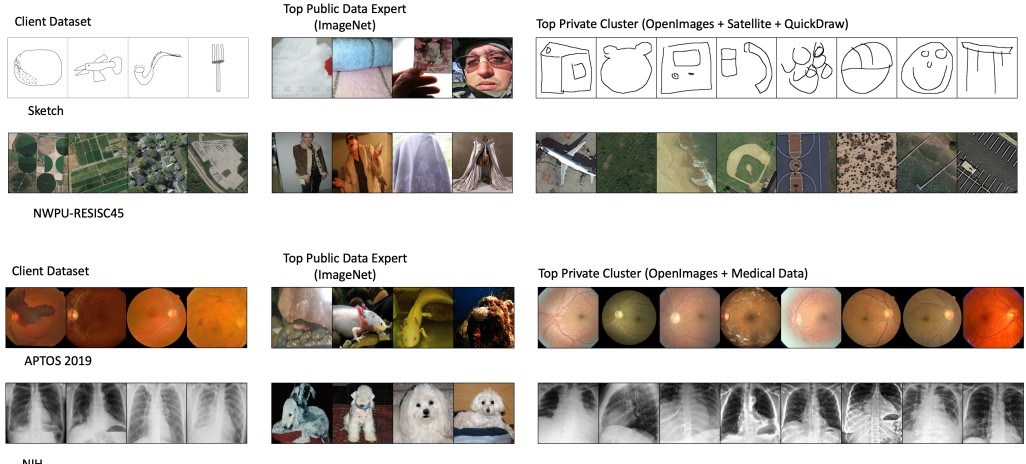

Figure 8: From left to right: Images from client/consumer dataset, images from "most similar" public split (ImageNet), Top cluster in corresponding mixed source dataset.