# OpenReview forum: "Scalable Neural Data Server: A Data Recommender for Transfer Learning"
_NeurIPS.cc/2021/Conference — NeurIPS 2021 Poster_

### Official Review · Reviewer_2egD · 2021-07-03

**Rating:** 7
**Confidence:** 3

**Summary:**

This paper presents an idea of a data recommender system called Scalable Neural Data Server (SNDS) to support transfer learning for target customers. The interesting point is that SNDS does not store any source or target dataset on the server, but only maintains the benchmark intermediary datasets, and the pools of experts that are trained on the intermediary datasets. SNDS shares the pool of experts to both data providers and consumers to compute similarities between the consumer's data and the data provider's datasets. Image rotation prediction is used to measure the expert model's performance on the source and target datasets. In this way, SNDS indirectly measures the data similarity between the sources and targets. Compared to its predecessor NDS, SNDS trains experts for an intermediary public dataset, instead of training an expert for each source. Using this method, SNDS achieves scalability as well as securing datasets from both sides. Extensive experiments are done to gauge the usefulness of data recommended by SNDS. With SNDS-recommended source, classification performance on target datasets improved significantly. To support this, the authors show that domain confusion exists between target datasets and recommended sources. The performance gain using SNDS is comparable to that of NDS, even though SNDS trains experts on the public datasets and promises cheaper computation costs. Besides, SNDS generalize to types of images that are not represented in the public dataset.

**Ethical Concerns:**

There are no ethical concerns. The reviewer can guess that this work is done by the authors of the neural data server (CVPR 2020) due to the similar style of Figure 1 and some figures in the experiments section. This evidence alone does not break anonymity, so it is good to go.

**Limitations And Societal Impact:**

Limitations and societal impacts are mentioned in the manuscript. As mentioned in the paper, providing a data approval process to block uploading inappropriate datasets can be needed to mitigate the potential negative impact of the data-sharing platform.

SNDS can be further improved for some domains such as medical data. From Figure 6, I felt SNDS recommends relevant images on source datasets. Was the problem that the performance didn't increase?



**Main Review:**

Recommending datasets without having it seems like a great innovation to me who came across the Neural Data Server concept through this paper. But, reviewers who already knew of NDS could look at this as incremental work. Though there is a huge concept change between the two works and it is worth noticing. The major change is not to provide experts trained from source datasets to clients. By using expert models trained from intermediary datasets, SNDS can guarantee extra anonymity between the customers and data providers. For me, it is not very clear why this solution scales cheaply with the number of sources. In Appendix D, since the first term (K x n x iter time) dominates the second term (K x M x match time), does SNDS is more efficient than [44] if M (# of data providers) >> K (public splits)?

Just wanted to clarify these again:
- Does the number of split K mean how many clusters the public dataset is divided into through k-means?
- Does the expert model is a model to classify the rotation amount of each partition?

Then, here is my question. Appendix F shows the performance change by reducing the number of splits. For some target datasets, the performance decreases. But overall, it increases on average. By connecting this observation to the computation cost difference between NDS and SNDS (M and  K), can you say it is okay to split the public dataset up to the number of source datasets (considering the match time is negligible)? Also, Have you tried a theoretical approach to prove a certain number of splits (K) is needed to approximate the source and target similarity when the number of source datasets is M?

Additional questions and typos:
- Should you denote \sigma_{i=1}^K ? instead of \sigma_{i}^K ?
- In Table 1, NDS is mentioned as an oracle probably because it provides experts trained on source data. But, why SNDS sometimes perform better than NDS? Would like to see standard deviation together with the averaged results.
- In Table 2, why SNDS (no target dataset) somehow outperforms SNDS in the 5% scenario?
- Figure 6: In the caption, Satellite data?
- Page 9: does can
- Page 13: Are you planning to open-source SNDS, or open it in commercial form?

Overall, I enjoyed reading this manuscript. The paper is well-written and the suggested framework is backed up with extensive experiments.

**Time Spent Reviewing:**

8

---

> ### Author Response · Authors · 2021-08-11
> **Reply to 2egD**
>
> Thank you very much for the review!
>
> Summary:
>
> - We clarify the terms regarding the computational efficiency of the method
> - We report results over 3 seeds for SNDS and NDS for the first 3 datasets (CUB200, Flowers102, Pets) in Table 1 (due to computational constraints)
> - We investigate the improvement of SNDS over NDS on certain tasks, and ultimately find that NDS is not a strict upper-bound over SNDS
> - We plan on open-sourcing our code upon acceptance
>
> 1. *In Appendix D, since the first term (K x n x iter time) dominates the second term (K x M x match time), does SNDS is more efficient than [44] if M (# of data providers) >> K (public splits)?*
>    As you mentioned, the computational costs per query are:
>    [44] $M \times n \times \text{iter time} + M \times \text{match time}$​​​
>    SNDS: $K \times n \times \text{iter time} + K \times M \times \text{match time}$​​
>    For both [44] and SNDS, the first terms dominate the second terms significantly, ie. computational costs per query is approximately:
>    [44] ~ $M \times n \times \text{iter time}$
>    SNDS ~ $K \times n \times \text{iter time} $.
>    When $M >> K$, the computational costs per query are greatly reduced for SNDS, due to the dominance of this term.
>
> 2. *Does the number of split K mean how many clusters the public dataset is divided into through k-means?*
>    Yes.
>
> 3. *Does the expert model is a model to classify the rotation amount of each partition?*
>    Yes, it is trained on the task of image rotation prediction.
>
> 4. *Also, Have you tried a theoretical approach to prove a certain number of splits (K) is needed to approximate the source and target similarity when the number of source datasets is M?*
>    We have not, however, this may be an interesting future direction for more theoretical work.
>
> 5. *Should you denote \sigma_{i=1}^K ? instead of \sigma_{i}^K ?*
>    Thank you for catching this typo, we will correct this in the paper.
>
> 6. *In Table 1, NDS is mentioned as an oracle probably because it provides experts trained on source data. But, why SNDS sometimes perform better than NDS? Would like to see standard deviation together with the averaged results.x*.
>    Due to the expensive computational cost of running the individual experiments, we have only obtained results averaged over 3 seeds for the first 3 datasets in Table 1 of the paper’s main body.
>    In this table, we show the mean and standard deviation over 3 runs with different seeds for SNDS and NDS. This corresponds to 3 different samples of data at each % of images for the selection methods.
>
>    | Selection Method | % of Images | CUB200       | Flowers102   | Pets         |
>    | ---------------- | ----------- | :----------- | ------------ | ------------ |
>    |                  |             |              |              |              |
>    | SNDS             | 2% (292K)   | 50.24 ± 0.88 | 82.66 ± 0.79 | 70.94 ± 1.17 |
>    | NDS              | 2% (292K)   | 49.95 ± 0.28 | 79.67 ± 0.80 | 70.85 ± 0.57 |
>    |                  |             |              |              |              |
>    | SNDS             | 5% (730K)   | 57.67 ± 1.05 | 87.02 ± 0.29 | 75.42 ± 0.74 |
>    | NDS              | 5% (730K)   | 58.33 ± 0.61 | 86.15 ± 0.13 | 75.36 ± 0.57 |
>    |                  |             |              |              |              |
>    | SNDS             | 10% (1.46M) | 60.81 ± 0.61 | 89.94 ± 0.52 | 77.09 ± 0.17 |
>    | NDS              | 10% (1.46M) | 60.23 ± 1.59 | 88.86 ± 1.77 | 77.35 ± 0.96 |
>
> 7. *In Table 2, why SNDS (no target dataset) somehow outperforms SNDS in the 5% scenario?*
>    We thank the reviewers for bringing this to our attention. We investigated this issue further, and found that SNDS and NDS occasionally differ significantly in their rankings of the source datasets, which can be beneficial to downstream performance. For example, we found that the top OpenImages clusters for the Flowers dataset differed between SNDS and NDS, and SNDS actually outperformed NDS on this dataset.
>    However, both SNDS and NDS recommend relevant clusters to the downstream task qualitatively in these cases (the majority classes appear to be relevant to the downstream task). For example, in the Flowers dataset, both of the top selected clusters contain Flower as the majority class (though these were grouped differently by the *K*-means algorithm). This leads us to believe that in some cases, the mechanism by which data selection is performed in SNDS can outperform NDS, despite not having an expert trained specifically for that cluster.
>    Hence, we will rename NDS from “oracle” in the final paper, since NDS is not a strict upper bound on SNDS performance.
>
> 8. *Figure 6: In the caption, Satellite data?*
>    Thank you for pointing out the typo, should state “medical data”.
>
> 9. *Page 9: does can.*
>    This is a typo, should only state “can”.
>
> 10. *Page 13: Are you planning to open-source SNDS, or open it in commercial form?*
>     We are planning to make SNDS available to everyone, by making the code open-source.

---

> > ### Comment · Reviewer_2egD · 2021-08-12
> > **Thanks for your responses.**
> >
> > Things that I did not understand are cleared up with the responses. Thanks!

---

### Official Review · Reviewer_yKh5 · 2021-07-08

**Rating:** 6
**Confidence:** 3

**Summary:**

The paper presents a scalable approach to recommend datasets from data providers to consumers looking to assemble datasets to pre-train models for downstream tasks. The approach does not leak source data from data providers and target data from consumers, and instead uses an intermediary public dataset and expert models trained on subsets of this public dataset to evaluate the data in sources that are likely to benefit the target tasks. The approach is scalable (independent of the number of source datasets), does not transfer source or target datasets to a server, transfers only expert models, and evaluates these expert models only on the source datasets and target datasets in the provider and consumer servers alone to keep computation costs low. The experiment results demonstrate the efficacy of this method, and show (somewhat surprisingly) that it is able to identify relevant source datasets even when the intermediate public dataset does not contain data relevant to the target downstream tasks.

**Ethical Concerns:**

As the authors say, malicious actors can manipulate the source datasets. This risk can be mitigated by a review system.


**Limitations And Societal Impact:**

The paper proposes the scalable data recommendation method for the task of image classification. From the paper, it is not clear how well this approach generalizes to other domains (like NLP, time series, speech/audio etc.) and other task (like object detection, segmentation etc.). I suspect it can generalize to any domain/task that uses pre-trained models, but this is neither claimed or dismissed in the paper.

As the authors say, the approach can benefit society by making it easier for consumers to identify source datasets likely to benefit their downstream tasks, without having to exchange data a-priori with the provider.



**Main Review:**

Originality: The paper extends an earlier method - neural data server to recommend datasets for pre-training models for downstream tasks, making it more scalable and secure for consumers. It relies on the fact that self-supervised training of experts on intermediate public datasets on a generic task (predicting rotation of images) captures some representations of images, and that these experts can find datasets with similar representations in source (from data providers) and targets (consumers who want to assemble pre-training data). While the idea is relatively common, the application to scale up the neural data server is novel and appears to be original.

Quality: The paper motivates the problem well, explains the solution approach, demonstrates the results using reasonable baselines and experiments, and addresses the limitations. The tests are rigorous (within the computational limits the authors mention) and results are convincing.

Clarity: I think the paper can use one round of clarifying edits. For example:
1)  it took a while to even realize that the authors focus on vision tasks.
2) sec 3.2: claim that image rotation prediction is correlated with domain confusion can use some references of related papers
3)  sec 3.3:  what if the source dataset happens to be rotated versions of the public dataset?
4) sec 4.3.1: I guess the section refers to Table 1, but nowhere is this mentioned
5) sec 4.3.1: In Table 2, Why do some metrics go up for SNDS compared to NDS at 5% sampling? What metrics are shows in Table 2?
6) Fig 5. left panel. Where is the result for Ngiam et al (green line)?
7) Table 3 - is the difference in metrics for NWPU-RESISC45 dataset significant?

Significance: With the growing availability of source datasets from providers who are however not willing to share the datasets publicly, and the increased use of pretrained models on large datasets for downstream tasks through transfer learning (or fine tuning), this approach offers a scalable way to identify specific source dataset relevant to a target downstream task, thus establishing a marketplace that helps both providers and consumers understand the value of a dataset without having to share it a-priori. This is a significant problem, the solution of which can open up broader applicability of machine learning.


**Time Spent Reviewing:**

4 hours

---

> ### Author Response · Authors · 2021-08-11
> **Reply to yKh5**
>
> Thank you for the review!
>
> Summary:
>
> - Our claim regarding image rotation prediction being correlated with domain confusion is supported by previous work
> - We believe that the proxy task can be substitutable to be rotationally invariant, as needed
> - We investigate the improvement of SNDS over NDS on certain tasks, and ultimately find that NDS is not a strict upper-bound over SNDS
> - We find a noticeable improvement in the NWPU-RESISC45 dataset over the baseline, even in this dissimilar setting
>
> More detailed answers are below:
>
> 1. *it took a while to even realize that the authors focus on vision tasks.*
>    Thank you for the feedback, we will clarify that the server is currently focused on computer vision tasks and datasets in all our experiments.
>
> 2. *sec 3.2: claim that image rotation prediction is correlated with domain confusion can use some references of related papers.*
>    This claim is from NDS [44], see Figure 4 in that work.
>
> 3. *sec 3.3: what if the source dataset happens to be rotated versions of the public dataset?*
>    This is an interesting suggestion that we have not tried, as we followed the setting in past work (NDS) and focused on testing SDNS with a variety of realistic image datasets. However, we believe that the proxy task used (unsupervised rotation) can easily be replaced with any of the more recent self-supervised methods (BYOL, MoCo, etc), which would be less susceptible to rotation transformations.
>
> 4. *sec 4.3.1: I guess the section refers to Table 1, but nowhere is this mentioned*.
>    We will mention Table 1 in this section as suggested.
>
> 5. *sec 4.3.1: In Table 2, Why do some metrics go up for SNDS compared to NDS at 5% sampling? What metrics are shows in Table 2?*.
>    We thank the reviewers for bringing this to our attention. We investigated this issue further, and found that SNDS and NDS occasionally differ significantly in their rankings of the private datasets, which can be beneficial to downstream performance. For example, we found that the top OpenImages clusters for the Flowers dataset differed between SNDS and NDS, and SNDS actually outperformed NDS on this dataset.
>    However, both SNDS and NDS recommend relevant clusters to the downstream task qualitatively in these cases (the majority classes appear to be relevant to the downstream task). For example, in the Flowers dataset, both of the top selected source clusters contain Flower as the majority class (though these were grouped differently by the *K*-means algorithm). This leads us to believe that in some cases, the mechanism by which data selection is performed in SNDS can outperform NDS, despite not having an expert trained specifically for that cluster. Following your advice, we have remarked in the paper that NDS is not a strict upper bound on SNDS, and provide the above experiment in supplementary material.
>
> 6. *Fig 5. left panel. Where is the result for Ngiam et al (green line)?*
>    Ngiam et al has a value ~1 in our simulation in the left curve (Computational Cost Per Query with Growing Server), so it appears as a flat line near 0.
>
> 7. *Table 3 - is the difference in metrics for NWPU-RESISC45 dataset significant?*.
>    We believe that it is fairly noticeable for such a dissimilar setting such as satellite imagery. This demonstrates that SNDS is able to recommend relevant data even outside of natural images. However, we believe that the difference in performance benefit on this dataset is less pronounced than in the Quickdraw dataset because of the task being more simple and prone to overfitting. This is emphasized by the authors of the original paper [11] using a 10%-90% train/test split, rather than a traditional split where the test set is much smaller than the train set.

---

### Official Review · Reviewer_eRqo · 2021-07-17

**Rating:** 5
**Confidence:** 3

**Summary:**

This paper extends Neural Data Server (NDS), which recommends relevant data points for the target task, to be more scalable, by employing a fleet of a mixture of experts. Each expert is trained on data slices, which is based on predefined assumptions, and generates representations that will be used to select relevant data slices for a given task.

**Limitations And Societal Impact:**

No suggestions.

**Main Review:**

First of all, this paper fits more appropriate to distributed systems or data mining conferences than a ML conference.

**Originality** This paper extends a prior work, NDS, by training experts for the subsets of datasets rather than training experts per each data source entirely. Each data source is partitioned by a predefined metric. When the actual task is given, the consumer pipeline will select the subsets by computing cosine similarities using the output of each expert from each slice. Hence, the whole system has some innovation in constructing the distributed system; however, I do not see a clear novelty for the machine learning community.

**Quality** Few questions arose after reading this paper. First, each expert needs to be trained with prior knowledge (or hypothesis) for the upcoming task. Combined with the simple, not adaptive similarity metric to select the subsets, the system seems to be limited for only a few tasks that are very similar to the original task. The generalizability of the proposed algorithm is quite limited in that sense. Second, similarly, I am a bit confused about the claim at Line 200. Assuming the source task is the same as the target task is a bit too strong an assumption. What’s the point of having a large system if we can only subset the dataset based on the known assumptions. Can we just train multiple experts over the whole dataset and ensemble them if we already have knowledge of the target task? I may have missed something, but I am not fully convinced of this setup. Third, the gap between the random sample and the proposed method seems to be a bit too thin, and it is essentially the only baseline. Authors should have considered some other baselines such as simple cosine similarity based subset selection (since they are using cosine similarities to the output of experts).

**Clarity** The paper was mostly easy to follow and clear.

**Significance** As mentioned earlier, the machine learning contribution of this paper is not substantial.

Other comment:
There are multiple other selection algorithms ([1, 2] for example -- just did the Scholar search) that can find the most relevant data points. Authors may want to add discussion to the work.

[1] Bengio, Yoshua, et al. "Curriculum learning." Proceedings of the 26th annual international conference on machine learning. 2009.
[2] Bruno, Andreis, et al. "Stochastic Subset Selection for Efficient Training and Inference of Neural Networks." (2020).


**Time Spent Reviewing:**

7

---

> ### Author Response · Authors · 2021-08-11
> **Reply to eRqo**
>
> Thank you for the review! We will address your feedback in the order that it appears.
>
> Summary:
>
> - We believe our work is suitable for a machine learning conference, since we are tackling a transfer learning problem at its core
> - We do not require a-priori knowledge of the tasks from the client
> - The source and target tasks do not need to be the same, and we will edit Section 3.6 to make this clearer
> - An ensemble method over the distributions is not a viable solutions because SNDS is task-agnostic and recommends data
> - We do not include a cosine embedding based baseline because of the possibility of reconstructing images from a feature embedding, violating D3
>
> We respectfully disagree with the view that our paper is more suitable for a distributed systems or data mining conference. Datasets are an integral component of any machine learning pipeline, and as such, an examination of data selection is an important topic for machine learning researchers. Fundamentally, SNDS tackles optimal data selection for pretraining, which is a transfer learning problem, not a systems problem. Furthermore, previous works for data recommendation such as [44] (NDS) and [3] (Task2Vec) and works that explored transfer learning in dissimilar settings to natural images such as [28],[35] were featured in similar venues such as CVPR, ICCV and NeurIPS.
>
> 1.  *First, each expert needs to be trained with prior knowledge (or hypothesis) for the upcoming task. Combined with the simple, not adaptive similarity metric to select the subsets, the system seems to be limited for only a few tasks that are very similar to the original task.*
>    We believe there is a misunderstanding from the reviewer side regarding this question. The experts are trained independently of the downstream tasks, and we use ImageNet subsets for the public experts due to their ubiquity in transfer learning. The experts are trained in an unsupervised manner for image rotation and do not require client labels, allowing them to be used for data selection for arbitrary computer vision tasks. We mention this within the Limitations section in 4.3.2.
>    However, due to this point of confusion, we will highlight the ability for SNDS to scale to arbitrary tasks more prominently.
>
> 2. *Second, similarly, I am a bit confused about the claim at Line 200. Assuming the source task is the same as the target task is a bit too strong an assumption.*
>    Thank you for bringing this to our attention. This assumption is only made for the theoretical analysis in 3.6, which is used as motivation for disentangling the public and source datasets. This assumption does not apply to SNDS as a whole, which does not require the source and target tasks to be the same. We will clarify this more carefully in the final version of the work.
>
> 3. *Can we just train multiple experts over the whole dataset and ensemble them if we already have knowledge of the target task?*
>    As highlighted previously in Q1. SNDS is designed to be agnostic to the labels/tasks of the output, and so the server provides data that can be used in a suitable manner for the client, rather than pretrained models that can only work for a specific task.
>
> 4. *Authors should have considered some other baselines such as simple cosine similarity based subset selection.*
>    We do not consider baselines that rely on computing a feature embedding from the source and client datasets due to Desiderata D3, which prohibits data from the data providers and client from being revealed during the process. This is due to the extensive literature on reconstructing images from feature vectors [49], [50], [51], [52], which would allow an attacker to reveal data about the client/data providers.
>
> *There are multiple other selection algorithms ([1, 2] for example -- just did the Scholar search) that can find the most relevant data points. Authors may want to add discussion to the work.*
> We thank the reviewer for bringing these works to our attention, though they do differ from our setting of a transfer learning and task-agnostic data recommendation problem. We will strongly consider adding these to our discussion section.
>
> [49] Mahendran, Aravindh and A. Vedaldi. “Understanding deep image representations by inverting them.” *2015 IEEE Conference on Computer Vision and Pattern Recognition (CVPR)* (2015): 5188-5196.
>
> [50] Dosovitskiy, A. and T. Brox. “Inverting Visual Representations with Convolutional Networks.” *2016 IEEE Conference on Computer Vision and Pattern Recognition (CVPR)* (2016): 4829-4837.
>
> [51] Yin, Hongxu et al. “Dreaming to Distill: Data-Free Knowledge Transfer via DeepInversion.” *2020 IEEE/CVF Conference on Computer Vision and Pattern Recognition (CVPR)* (2020): 8712-8721.
>
> [52] Dong, Xin et al. “Deep Neural Networks are Surprisingly Reversible: A Baseline for Zero-Shot Inversion.” *ArXiv* abs/2107.06304 (2021): n. pag.

---

> > ### Comment · Reviewer_eRqo · 2021-09-01
> > **Re: Reply to eRqo**
> >
> > Thanks for the detailed response. I partially agree few points that author claims, particularly the constraint of Desiderata D3, the privacy constraint. However, I do have some remaining concerns on the #1 point that I brought up.
> >
> > I am not trying to get to too extreme scenario; however, the selection of the experts still requires some prior assumption of the data. For example, if a SNDS system is using ImageNet as experts, it has a fundamental assumption that the end-goal task will be a vision task. Moreover, the images should be scale variant, rotation variant to the end-goal task since the experts are trained such ways. This vision SNDS system also *cannot* handle text data in its extreme.
> >
> > In order to setup a true universal task-agnostic system, SNDS requires infinite number of experts theoretically. For me, the scalability of SNDS actually comes from introducing **reasonable** scoping of the task assumption, which is practically sound solution. I hope authors can add discussions on this, in the future version of the paper.
> >
> > I read other reviewers comments and agree on the main contribution of the paper. I will increase the score accordingly.

---

### Author Response · Authors · 2021-08-11
**Overall Response**

Thank you to the reviewers for the comments regarding our work. In particular, we appreciate that reviewer **yKh5** noted “*The experiment results demonstrate the efficacy of this method, and show (somewhat surprisingly) that it is able to identify relevant source datasets even when the intermediate public dataset does not contain data relevant to the target downstream tasks*” and **2egD**: *“The performance gain using SNDS is comparable to that of NDS, even though SNDS trains experts on the public datasets and promises cheaper computation costs.”*

We believe that we have addressed the main concerns regarding our work: **1)** We have explained why our work is suitable for this conference and showed that similar works has been presented at similar venues **2)** We have emphasized that our method is task-agnostic and does not require a-priori knowledge of the task **3)** We have added additional experiments reporting averaged runs over a subset of our datasets from Table 1 **4)** We have clarified that we will release our code as open-source upon acceptance.

---

### Decision · Program_Chairs · 2021-09-27

**Decision:**

Accept (Poster)

**Comment:**

Quite a borderline paper with a high variance of scores. I'm recommending acceptance though the average score is probably below what would correspond to an accepted paper at NeurIPS. Mostly I'm siding with the positive reviewers because:
-- The negative score/review is somewhat of an outlier compared to the other two.
-- The two positive reviewers voiced their support of acceptance in the discussion, whereas the negative reviewer did not weigh in. This caused me to downweight their assessment a little.
-- The rebuttal seems reasonable, and was at least persuasive to the most positive reviewer. Again, the negative reviewer was rebutted though did not respond (which I take as somewhat implicit support).

Ultimately a tough call. The positive reviewers found the problem important, the solution effective, and praised the introduction of new datasets for the task. The negative reviewer criticized the novelty of the work, though again this was not a consensus view. Most reviewers raised issues of clarity, though these can likely be addressed in a revision (or have already been addressed in the rebuttal).

Still may be borderline given that the paper doesn't quite have a "champion" and likely will have lower scores than other accepted papers. But there is reasonable endorsement by the reviewers (especially weighted by participation in the discussion), and the criticisms seem like not major red flags.